# Semisolid Wet Sol–Gel Silica/Hydroxypropyl Methyl Cellulose Formulation for Slow Release of Serpin B3 Promotes Wound Healing In Vivo

**DOI:** 10.3390/pharmaceutics14091944

**Published:** 2022-09-14

**Authors:** Mattia Albiero, Alice Fullin, Gianmarco Villano, Alessandra Biasiolo, Santina Quarta, Simone Bernardotto, Cristian Turato, Mariagrazia Ruvoletto, Gian Paolo Fadini, Patrizia Pontisso, Margherita Morpurgo

**Affiliations:** 1Department of Medicine, The University of Padova, Via Giustiniani 2, 35128 Padova, Italy; 2Veneto Institute of Molecular Medicine, Via G. Orus 2, 35129 Padova, Italy; 3Department of Pharmaceutical and Pharmacological Sciences, The University of Padova, Via Marzolo, 5, 35131 Padova, Italy; 4Department of Surgical, Oncological and Gastroenterological Sciences-DISCOG, University of Padova, Via Giustiniani 2, 35128 Padova, Italy; 5Department of Molecular Medicine, University of Pavia, 27100 Pavia, Italy

**Keywords:** sol–gel silica, protein formulation, semisolid formulations, serpin B3, diabetic ulcer, chronic ulcer, wound healing

## Abstract

Foot ulcerations are a disabling complication of diabetes and no treatment is currently available based on disease mechanisms. The protein serpin B3 (SB3) was identified as a positive biomarker of successful diabetic wound healing; therefore, its exogenous administration may promote healing. The topical administration of SB3 is challenging due to its protein nature. Physical entrapment in wet sol–gel silica can stabilize the protein’s conformation and permit its sustained delivery. However, irreversible syneresis and poor viscoelastic properties hamper wet sol–gel silica application as a semisolid vehicle. To overcome these limits, a sol–gel silica/hydroxypropylmethylcellulose (HPMC) hydrogel blend was developed. SB3 entrapped in 8% SiO_2_ wet sol–gel silica preserved its structure, was stabilized against denaturation, and was slowly released for at least three days. Blending a silica gel with an HPMC–glycerol (metolose-G) hydrogel permitted spreadability without affecting the protein’s release kinetics. When administered in vivo, SB3 in silica/metolose-G—but not in solution or in metolose-G alone—accelerated wound healing in SB3 knockout and diabetic mouse models. The results confirmed that SB3 is a new pharmacological option for the treatment of chronic ulcers, especially when formulated in a slow-releasing vehicle. Silica–metolose-G represents a novel type of semisolid dosage form which could also be applied for the formulation of other bioactive proteins.

## 1. Introduction

Within their lifetime, about 15% of patients with diabetes mellitus are bound to develop foot ulcerations, a chronic condition of delayed healing which often leads to amputation, thereby causing the patient to be burdened with significant morbidity and mortality. Indeed, diabetes is the leading cause of non-traumatic limb amputation in western countries, and the mortality after a major amputation is similar to or worse than that of several malignant cancers [1]. Wound healing is a complex process involving several tissues and cell types, and is characterized by different and coordinated phases, including inflammation, coagulation, fibrogenesis, the formation of the granulation tissue through angiogenesis, fibrinogenesis, and its regression [2]. Any alterations to the above-described events can delay healing and lead to wound complications.

The mechanisms responsible for the delayed healing of ulcers in diabetes are manifold and not yet fully understood [3]. In a recent proteomic study carried out on ulcer tissue samples from patients with diabetes, the serpin B3 (SB3) protein was identified as a positive biomarker of successful healing. Serpin B3 is a protease inhibitor involved in the activation of fibrogenesis and angiogenesis, which are two key processes of wound resolution [4,5]. SB3 was strongly downregulated in non-healing diabetic wounds as compared to rapidly healing wounds, and such a finding was validated using proteins and mRNA from individual samples [6,7]. These findings prompted us to investigate whether the local administration of exogenous SB3 can improve diabetic wound healing. However, the topical administration of SB3 at an open ulcer presents some technical challenges due to the protein nature of this agent. In fact, the ulcer environment is normally aggressive to proteins (which are prone to inactivation secondary to proteolytic digestion and/or physical denaturation) and could quickly reduce local SB3 levels, unless a constant supply of functional protein is provided. In order to achieve this, a prolonged release of the bioactive agent to the site of action is desirable. In addition, the viscoelastic properties of the formulation should permit a long-time residence at the application point, without dripping or drying out. Finally, the vehicle should be capable of preserving the active agent’s stability over time to guarantee the shelf life.

Although the administration of proteins at the outer skin may be useful for some pathologies, to date, little effort has been devoted to the development of specific protein formulations that are suitable for this route [8,9,10]. Indeed, most of the strategies for the delivery of proteins described in the literature are finalized to parenteral applications or local administration for the purpose of a transdermal or systemic effect [11]. On the other hand, many of the solutions optimized for parenteral uses (liposomes, micelles, or inclusion in hydrogels) may be suitable for external application, provided that the release characteristics of the active element and the viscoelastic properties of the formulation administered are suitable for this use.

In particular, the use of hydrogel carriers may fit this purpose. Hydrogels (which can be obtained using a wide variety of polymeric materials) can both stabilize proteins and generate systems with adequate viscoelastic properties to permit physical retention at the site of administration [12,13,14]. However, not many hydrogels are able to generate a slow and prolonged release of proteins, a property that is fundamental in order to provide a constant supply to the application site.

One possible solution may come from wet sol–gel silica-based vehicles [15]. Sol–gel silica is a highly porous inorganic polymer obtained synthetically in mild conditions from silicon alkoxide precursors. Sol–gel silica polymerization occurs in aqueous/alcoholic environments, which, in some cases, are compatible with the stability of proteins [16,17,18]. If the polymerization is carried out in the presence of a bioactive agent, the agent remains physically trapped within the newly generated polymer network.

The product of polymerization is a ‘wet’ monolith in which the silica network entrapping the bioactive agent is surrounded by an aqueous/alcoholic solution, the composition of which depends on the type and concentration of the reagents used. The initial gel can then be dried (xerogel or aerogel) or it can be used in the wet form [18].

In the case of proteins, it has been shown that when they are embedded in wet sol–gel silica, they often maintain their folded state, they are protected from denaturation [18,19,20,21], and they can be slowly released (within days or weeks) secondary to diffusion and/or polymer erosion. Erosion, which occurs in aqueous environments, releases silicic acid—a component of mineral waters—which is biocompatible and nontoxic.

On the downside, sol–gel silica is not spreadable, and when exposed to air, it undergoes an irreversible syneresis, which reduces the polymer pore size and, thus, the speed of erosion and the diffusivity of the embedded proteins.

In order to overcome these limits, we investigated the possibility of blending a protein-loaded wet sol–gel silica with an organic hydrogel carrier, with the final goal of developing a formulation for the sustained delivery of SB3 to chronic ulcers.

Initially, we evaluated the effects of silica entrapment on SB3’s structure, biological activity, and release kinetics. A composite preparation suitable for topical application was then optimized by first crushing the wet sol–gel silica monolith into a wet gel slurry and then suspending it in a hydroxypropyl methyl cellulose (HPMC)-based hydrogel (Figure 1). The effects of both crushing and HPMC blending on the protein-releasing properties of the silica gel were investigated. Following the pre-formulation studies, an optimized SB3-loaded formulation was then generated, and its potential for treating wounds was compared to that of SB3 in a buffer or in the HMPC hydrogel using two animal models characterized by impaired wound healing: SB3 knockout (KO) mice and streptozotocin (STZ)-induced diabetic mice.

## 2. Materials and Methods

### 2.1. Materials and Instrumentation

Human (h)SB3 was obtained in our laboratory using recombinant molecular techniques, as previously described [22]. Mouse serpin B3 analogue (m)SB3 was purchased from NZYTech (Lisboa, Portugal); human anti-SB3 rabbit polyclonal Abs were kindly provided by Xeptagen S.p.A, (Vega Park, Venice, Italy); hydroxypropyl methyl cellulose (HPMC, code 90-SH-100000 SR) was obtained from Seppic (Milan, Italy); Kollidon 30 (# 63-0137) was obtained from BASF (Germany); pronase from *Streptomyces griseus* (# 2555112) was obtained from Roche Diagnostics; ECL Prime Western blotting detection reagent (# 9995053) was obtained from GE Healthcare; and bovine serum albumin (BSA # 078K0710), tetramethyl orthosilicate (TMOS, # BCBN3603V), fluorescein isothiocyanate (FITC # 039K5310), electrophoresis marker C1992-1VL (# 108K6401, MW: 8000-222,000), and all other reagents were from Sigma Aldrich (Merck, St. Louis, MO, USA).

BSA–FITC was obtained through the conjugation of BSA with 10 equivalents of FITC in PBS. The product (FITC:BSA = 2:1 mol:mol) was purified by extensive dialysis (10 kDa cut-off) against PBS, sterile filtered, and maintained at 4 °C until use. Spectroscopic analyses were performed with a Varian Cary 50 UV-Vis spectrophotometer. The fluorescence was measured using a JASCO FP-6200 spectrofluorometer. Circular dichroism spectra were recorded using a JASCO J-810 spectropolarimeter. Protein release studies were conducted using 1 mL Float-A-Lyzer G2 dialysis tubes from Spectrum Laboratories, Inc. A Hettich Universal 320-R centrifuge was used for the gel washes. A Timo autoclave was used for sterilization. A Memmert GmbH, model SV 1422, was used as a thermostated bath. Electrophoresis on a polyacrylamide gel was conducted using a gel casting from the Bio-Rad Mini-Protean System. Western blots were performed on Millipore PVDF (polyvinylidene fluoride) membranes. The Molecular Imager VersaDoc MP4000 was used for the acquisitions of the Western blot membranes.

### 2.2. Preparation of Wet Sol–Gel Silica Gel Slurries

All steps were performed in a sterile environment to prevent microbial contamination. Gel monoliths (8% SiO_2_ *w*/*v*) were obtained from tetramethoxysilane (TMOS) according to a procedure already described [18,20]. Briefly, the partial hydrolysis of the alkoxysilane precursor (activated sol) was initially promoted with an acidic treatment at room temperature for 30 min using HCl (Si:HCl:H_2_O molar ratios = 1:6 × 10^−6^:1.25). For each mL of wet sol–gel silica, 230.4 µL of activated sol was mixed, under a gentle vortex, with 769.6 µL of protein solution in PBS + 0.1% Kollidon 30. When used as such, the monoliths were rinsed (3×) with 5 volumes of buffer (100 mM phosphate, pH 6.0) to extract the methanol liberated during condensation. The process allowed for the quantitative entrapment of the protein in solution. Wet gels were prepared at a protein load varying between 5 and 200 µg/mL of gel.

Wet gel slurries were obtained immediately after the monolith formation by manual crushing using a sterile glass rod, and were washed thoroughly (3×) by adding 5 volumes of buffer (100 mM phosphate, pH 6.0), centrifuging (1390× *g*, 1 min), and removing the supernatant.

The products were used immediately after preparation or stored at 4 °C and sealed with a minimum amount of buffer to prevent drying. For the in vivo administration, the gel slurry (200 µg of SB3/mL) was diluted to 1:200 into a sterile 3% HPMC hydrogel (metolose) containing 10% *w/v* glycerol (metolose-G).

### 2.3. Structural Conformation of SB3

The SB3’s protein conformation in solution and within the silica sol–gel was determined using circular dichroism. An analysis was carried out at 25 °C in a quartz cuvette with an optical path of 1 mm, and the following parameters were set: λ: 190–260 nm; bandwidth: 2 nm; response time: 16 s; sensitivity: standard; data pitch: 0.1 nm; scanning speed: 10 nm/min; and acquisitions: 2 (for blank) and 4 (for protein). The spectra in solution were registered at 0.15 mg/mL of SB3 in 100 mM phosphate with a pH of 6.0. The spectra in the wet gel (SB3 at 0.3 mg/mL of gel) were registered from the wet silica slurry suspended in an equal volume of buffer inside the quartz cuvette.

### 2.4. SB3 Thermal Stability

The protein’s t-melt in solution (0.15 mg/mL in 100 mM phosphate, pH 6.0) and when trapped within the wet gel matrix (0.3 mg/mL in the gel, diluted 1:1 with 100 mM phosphate, pH 6.0, for suspension) was determined by circular dichroism. Repeated CD spectra were acquired (between 200 and 260 nm with 0.1 nm intervals) at 10 °C intervals in a temperature range between 40 °C and 90 °C, with a heating rate of 60 °C/h. The thermal denaturation curves were obtained by plotting the mean ellipticity value per residue, measured at 222 nm as a function of the temperature of analysis.

### 2.5. SB3 Stability to Proteolysis

SB3 was analyzed by Western blot (after 12% SDS PAGE) after its release from a wet sol–gel silica slurry previously incubated with the proteolytic enzyme pronase. More into details, prior to release, the wet sol–gel silica gel slurry (200 µL) containing SB3 (300 µg/mL of gel) was suspended in 500 µL of pronase (3 µg/mL; SB3:pronase *w/w* ratio = 40:1) in HEPES (10 mM) + NaCl (150 mM) + CaCl_2_ (10 mM), with a pH of 7.4. After 1 h of incubation at 37 °C, the gel was washed 6 times with 100 mM phosphate with a pH of 6.0 to remove the pronase and was then incubated at 37 °C in 10 mL of release buffer (HEPES buffer (100 mM) + 0.1% BSA + 0.05% Tween 20, pH 7.4). After 1 h, the release solution was analyzed using SDS-PAGE and Western blot. The SB3 on the membrane was detected with ECL after incubation with rabbit anti-human SB3 antibodies, followed by goat anti-rabbit IgG HRP. Similarly, the protein in solution was incubated with pronase at an SB3:pronase *w/w* ratio = 40:1 for 1 h at 37 °C and then analyzed by WB. As a control, the same release protocol was carried out on an SB3–gel slurry that had not been treated with pronase.

### 2.6. Protein Release from Wet Silica Gel

The wet silica gel (200 µL of either monolith or slurry) was suspended at 37 °C in 10 mL of 100 mM phosphate buffer, 0.05% Tween 20, and 0.1% BSA at a pH of 6.0 and incubated at 37 °C under gentle shaking. At defined time points (0 h, 5 h, and 24 h), a small portion of the supernatant (250 µL) was removed for SB3 titration and replaced with fresh buffer. The gel was then re-suspended in the supernatant using a gentle vortex and the incubation was continued until the following time point. When testing the gel slurry, the sample was centrifuged (1390× *g*, 3 min) before supernatant isolation.

In order to evaluate the influence of HPMC hydrogel blending on the protein release from wet sol–gel silica, slurries embedded with BSA–FITC were suspended in an equal volume of release buffer 3% HPMC, inserted into dialysis tubes (cut-off of 100 kDa), and immersed in PBST + 0.1% BSA at 37 °C. At scheduled time points, the protein content in the acceptor solution was quantified.

BSA–FITC titration in the release medium was carried out using fluorescence spectroscopy. SB3 titration was performed with ELISA testing (HEPA Lisa kit, Xeptagen S.p.A, Vega Park, Venice, Italy) by following the manufacturer’s instructions. Briefly, the release solutions obtained from the gels loaded with SB3 at 5 or 50 µg/mL were diluted to 1:5 or 1:50, respectively, and were incubated for 1 h at room temperature on 96-well plates coated with rabbit anti-human SB3 capture Abs (10 μg/mL in PBS, pH 7.4). A standard curve, obtained through the dilution of recombinant SB3 from 16 to 0.25 ng/mL, was also included as a calibrator. After washing, SB3 was revealed through incubation with 100 μL of HRP-conjugated streptavidin secondary anti-SB3 Abs (0.5 μg/mL). The plate was developed with a ready-to-use 3,3′, 5,5′-tetramethylbenzidine (TMB) substrate solution. The reaction was stopped with 1 mol/L of HCl (100 μL) and the absorbance at 450 nm was measured on a microplate reader (Victor × 3; Perkin Elmer, Waltham, MA, USA). All samples were tested in duplicate. The results were graphically represented by plotting the ratio between the quantity of released protein at each time point (Mt) and the total protein (Mtot) embedded as a function of the time.

The activity of the released SB3 was verified with a scatter assay, as previously reported [23]. In detail, Madin–Darby canine kidney cells (MDCK) (5 × 10^3^ cells/well) were treated with two different dilutions of the release media (1:16 and 1:128) obtained after 0, 5, and 24 h of the release assay. Cells were fixed in 4% formaldehyde and stained with Coomassie blue solution, and the scattering effect was analyzed using a 20 Axiovert 200M microscope (Carl Zeiss Microscopy GmbH, Jena, Germany). As a control, positive recombinant SB3 was used at 200 ng/mL.

### 2.7. In Vivo Testing

Animal experiments were approved by the University of Animal Care Padua and the Italian Ministry of Health (Aut. No. 171/2019-PR). SB3 knockout mice [24] were bred in-house. For the diabetic model, diabetes was induced using a single intraperitoneal injection of 150 mg/kg of streptozotocin (STZ) (Sigma Aldrich) in citrate buffer (50 mmol/l, pH 4.5). The blood glucose was measured with a Glucocard G-meter (Menarini, Florence, Italy); mice with a blood glucose level ≥ 16.7 mmol/L (300 mg/dL) in at least two measurements within the first week were classified as diabetic and housed for 4 weeks with free access to food and water before the experiments were performed. To create skin wounds, mice were sedated through sodium isoflurane inhalation, and the skin of the back was shaved with Veet cream and disinfected with povidone iodine at 10%. Then, 4 mm-diameter wounds were inflicted with a biopsy punch (H-S Medical, Colton, CA, USA).

The animal wounds were treated locally with 100 ng of SB3 formulated in 100 μL of 100 mM phosphate buffer + 0.1% Kollidon-30, with a pH of 6.0, or 100 μL of metolose-G or 100 μL of metolose-G containing wet silica slurry (metolose-G:silica slurry = 200:1 *v*:*w*) (SB3 in wet silica gel = 200 μg/mL). Metolose-G and an SB3-free wet silica slurry diluted by 200× in metolose were used as controls. Each treatment (N ≥ 7 for each group) was administered twice, immediately after the execution of the wound and the day after. During the observation period, no other wound dressing was applied. To evaluate the time to wound closure, photographs were taken at selected time points, and the wound area was quantified with ImageJ.

### 2.8. Statistical Analysis

Statistical analyses were carried out using the Graph Pad software (v. 8.0.2, San Diego, CA, USA). Two-way ANOVA was used to assess significance, which was considered for *p*-values below 0.05.

## 3. Results

### 3.1. Wet Sol–Gel Silica Entrapment Maintains SB3 Conformation and Improves Its Stability

The protein-embedded wet sol–gel silica was prepared following a process already described [18] and summarized in Figure 1. Our intention was to treat the wounds with a single administration, so a gel capable of releasing the protein for at least 3–4 days was planned. Therefore, a SiO_2_ concentration in the wet gel equal to 8% (*w/v*) was selected, with the expectation—according to the literature data [20]—of about 30–50% release after 24 h.

#### 3.1.1. SB3 Conformation and Stability

The circular dichroism spectra of the protein in solution and in the silica gel slurry (Figure 2A) were superimposable and displayed two minima at 222 nm and 208 nm, which are typical of proteins with a prevalent α-helix conformation. This demonstrates that the polymerization process, followed by the washing steps, was mild to the protein, which remained in its folded state. This was not obvious, considering that the polymerization of the 8% SiO_2_ gel liberated 20% (*v/v*) of methanol as the result of tetramethoxysilane condensation, and the presence of this solvent may induce protein denaturation.

The stability of the protein in the buffer or in the wet sol–gel silica was also compared. The SB3 in the buffer was quite stable, as it maintained its conformation upon a thermal ramping up to almost 70 °C (Figure 2B). However, entrapment in the wet sol–gel silica made it even more stable: no change to its conformation was detected at all temperatures tested (up to 90 °C).

#### 3.1.2. SB3 Resistance to Proteolytic Attack

The stability of the protein against proteolysis, both in solution and in the gel, was measured. Proteolytic resistance is an important feature, since wound environments are rich in proteolytic enzymes, which may degrade the protein down to a loss of function [25]. Figure 2C displays the Western blot analysis of SB3 prior to and after treatment with pronase, a potent non-specific mixture of proteolytic enzymes from *Streptomyces griseus* K^−1^ [25]. The native SB3 displayed a single band at 45 kDa, in accordance with its molecular weight. Upon protease treatment, the 45 kDa band disappeared, demonstrating that, despite being a protease inhibitor, SB3 can be cleaved by the non-specific proteolytic enzyme mixture of pronase. A WB analysis was also carried out on the solutions obtained by performing a release assay from SB3-loaded wet silica gels as prepared, or those previously treated with pronase [25]. Notably, the silica gel release solution still displayed the 45 kDa band, the intensity of which was comparable to that obtained in the absence of pronase. The results demonstrate that, as long as SB3 is confined within the wet silica gel network, it is protected from proteolytic attack.

### 3.2. Embedment into Wet Sol–Gel Silica Permits Sustained Release of SB3 in a Functional Active Conformation

The silica slurry, when immersed in an aqueous environment, guaranteed a controlled and prolonged release of SB3 (Figure 3) and the proteins released were functional, as assessed by the scatter assay on MDK cells [23]. The release rate did not depend on the protein gel load, at least in the concentration range here investigated. The release was diffusion driven, since its rate was linear with the square root of time. About 30% of the release was reached after 24 h and <60% after 72 h. This rate, which is in line with literature data on other proteins embedded in similar wet sol–gel silica, should guarantee a supply of the active agent for at least 3 days [22].

### 3.3. Formulating Protein-Loaded Wet Sol–Gel Silica into a Vehicle Suitable for Topical Administration

To generate a wet sol–gel silica-based formulation suitable for topical administration, the SB3-containing wet gel monolith obtained after polymerization was manually crushed into a coarse gel slurry (e.g., using a glass rod or a rotating palette) and then suspended into an HPMC-based hydrogel. In principle, both the process of size reduction and blending with the organic polymer hydrogel may have affected the properties of the silica gel. Therefore, we verified if and to what extent each of these steps affected the protein release. In these experiments, BSA (which became fluorescent through conjugation with fluorescein isthiocyanate (FITC) into BSA–FITC) was used as a model protein instead of SB3. BSA is available in large quantities, is similar to SB3 in terms of its MW (69,293 vs. 45,832), and is negatively charged in a physiological buffer (isoelectric points are 5.82 vs. 6.8). Earlier studies have also shown that BSA’s conformation is not altered by wet gel silica entrapment [18], and therefore it is a good model for predicting the behavior of SB3 in similar circumstances.

#### 3.3.1. Effect of Manual Gel Crushing on Protein Release

The release profiles of BSA–FITC embedded in 8% SiO_2_ wet gels, either in the monolith shape or as a crushed gel slurry, were compared [15]. Both the gel slurry and the monolith displayed a diffusion-driven process (a release that was linear with the square root of time) (Figure 4). About 60% of the embedded protein was released after 24 h in both cases, even if, in the case of the monolith, the process was slightly slower in the early phase. The release registered for BSA was faster than the one observed for SB3 (Figure 2), and this difference may depend on the specific nature of the two proteins. However, the difference in the release rate between the two gel geometries was barely significant. This finding agrees with the literature [18,20], which states that, in the dimensional ranges here used, the size of the gel does not significantly influence the speed of silica erosion, which is the prevalent phenomenon responsible for protein release.

#### 3.3.2. Effect of HPMC Hydrogel on Protein Release from Wet Sol–Gel Silica

The suspension of the gel slurry into the HPMC hydrogel was investigated with the twofold objective of improving the formulation’s spreadability and preventing the syneresis of the wet silica network. A 3% *w/v* HPMC concentration was identified based on preliminary rheology tests (not shown), since this allowed the formulation to maintain a viscosity suitable for topical application (no dripping) even after autoclave sterilization. Glycerol (10% *w/v*) was also added to the 3% HPMC hydrogel with the aim of slowing down water evaporation (see Appendix A). The HPMC/glycerol 10% mixture (metolose-G) was very viscous; in principle, this may slow down the diffusion of the protein after it is released from the silica gel. We ruled out this possibility by testing formulations prepared with the model protein BSA–FITC. As shown in Figure 5, the release rates from the silica slurry in the presence or absence of the organic polymer hydrogel were almost superimposable, indicating that, despite its high viscosity, the HPMC hydrogel network did not affect protein mobility. Therefore, we can conclude that the only component in the wet silica–metolose-G formulation responsible for the protein’s slow release was the silica slurry.

### 3.4. Serpin B3 in Wet Silica–Metolose-G Is Effective at Restoring the Ulcer-Healing Process in Mouse Models

Wound healing is a complex, multi-step process that cannot be fully recapitulated in vitro. Therefore, animal models of wound healing are an established tool for studying this process in vivo, especially when associated with a pathophysiological setting such as diabetes, where impaired wound healing is still an unmet clinical need lacking a mechanistic treatment. In mice, wound healing occurs via contraction, rather than epithelialization [26], and surgical dressings or silicon splints can be applied to full-thickness wounds [27,28]. However, both strategies could interfere with the application of our formulation, and splints have been shown to reduce the differences in wound closure between diabetic and non-diabetic mice [28]. Thus, as shown in our previous work, we have chosen not to add any dressing or splints to the wounds [29,30].

Preliminarily to the in vivo studies, and to support the rationale for the use of murine SB3 (mSB3) in the animal models, the effect of SB3 in vitro on murine fibroblasts was verified (Appendix B).

#### 3.4.1. SB3 Knockout Mouse Model

To validate the wet gel silica slurry–metolose-G formulation, we initially used serpin B3 knockout (KO) mice. Indeed, this animal model, which has also been used to study other pathological states [24], is characterized by impaired wound healing (Appendix C). In a rescue experiment, serpin B3 KO mice underwent wounding and were treated with topical applications of freshly prepared mouse serpin B3 formulations in a wet gel silica slurry–metolose-G, metolose-G, or an aqueous buffer (Figure 6A). We found that serpin B3 delivered in the silica slurry–metolose-G was significantly more effective at improving healing at earlier time points compared to the other formulations. After 3 days, the ulcers treated with serpin B3 in silica–metolose-G had healed by 0.64 ± 0.09-fold of the basal area (mean ± SEM) compared to the ulcers where serpin B3 was delivered with either metolose-G or a buffer, which healed the wounds by 1.49 ± 0.11- and 0.99 ± 0.11-fold of the basal area, respectively (Figure 6B,C, *p* < 0.05).

#### 3.4.2. Diabetic Mouse Model

We then moved to a more clinically relevant model, which was streptozotocin (STZ)-induced diabetic mice. This model has been extensively used to study the adverse effects of high glucose on wound healing without the confounding factor of concomitant alterations associated with other risk factor such as dyslipidemia or hypercholesterolemia.

In a preliminary experiment (Appendix D), we confirmed that STZ diabetic mice displayed impaired would healing compared to non-diabetic controls, confirming that this model recapitulates the human disease in this aspect. In these preliminary experiments, we ruled out the possibility that the SB3-free vehicles of the different formulations exerted any significant effect on the healing process in diabetic or non-diabetic mice.

At 3 days after wounding (Figure 7), the animals treated with the wet gel silica slurry formulation showed faster healing, as the wound area was reduced to 26.9 ± 2.5% of the initial area (Figure 7C, *p* < 0.05 compared to untreated wounds). Collectively, our data show that SB3 prepared in the wet sol–gel silica slurry outperformed the other formulations for enhancing healing within the earlier days after wounding, which represents the most critical time for the proliferation of dermal fibroblasts to yield the granulation tissue [31].

## 4. Discussion

An ideal vehicle for the topical delivery of proteins to open wounds should be characterized by a number of properties: (a) it should not be harmful to the structure (and biological activity) of the protein and, possibly, preserve its stability upon storage to favor long-lasting efficacy and a long shelf life; (b) it should allow for the slow release of the active agent to guarantee a constant supply at the site of administration; and (c) it should possess proper viscoelastic properties to permit a long-time residence at the application point, without dripping or drying out.

The wet sol–gel silica here investigated proved to have ideal features in terms of protein stabilization and SB3 release kinetics [15,19]. Indeed, the SB3 entrapment into the 8% SiO_2_ silica gel did not alter its biological activity and allowed for its sustained release over at least 3 days, so that we could treat the animals with a single dose. Notably, since SB3 gel entrapment stabilized the protein against physical denaturation and proteolytic attack, the fraction of unreleased protein remained protected and active until released. Therefore, the silica gel acted as a stabilizing depot, providing a long-lasting supply of active protein along the entire release process.

An interesting result from the preliminary tests is that the shape and size of the silica gel did not affect the protein’s release kinetics. This may be explained, considering that the protein release from the wet sol–gel silica was mainly secondary to silica erosion, a phenomenon that is driven by water diffusing within the silica network. It is likely that, at the SiO_2_ concentration here used, the pores of the silica network were large enough not to impair the free diffusion of water molecules. As a consequence, silica erosion (and the protein release from it) occurred equally and at the same speed at both the periphery and from within the matrix. From a practical point of view, this result indicates that strict control over the size of the crushed particles is unnecessary.

The metolose-G blend used for suspending the sol–gel silica provided the textural properties necessary for topical application without affecting the protein release or stability. Both components were selected among others because of their extensive use in semisolid formulations for topical administration. Glycerol is a natural and highly biocompatible osmotic agent classically used as humectant and emollient in aqueous-based semisolid formulations at concentrations up to 30% *w/v* [32]. It is harmless to proteins when added to storing buffers at up to a 50% concentration, which is an important property when designing a formulation for protein drugs. HPMC was selected for its non-charged nature to avoid unwanted electrostatic interactions with the polyelectrolyte proteins. The 3% *w/v* HPMC concentration was used because it allowed for a viscosity suitable for topical application (no dripping) even after autoclave sterilization, which is a necessary process for formulations to be administered to open ulcers.

The addition of glycerol was fundamental to slow down water evaporation from the formulation, a phenomenon which is deleterious for wet silica as it yields an irreversible syneresis that stops the diffusion of the embedded proteins. Silica syneresis is a phenomenon that occurs when no liquid phase is left to surround the silica polymer network. In this respect, glycerol prevents syneresis primarily by slowing down water loss, as shown in Appendix A. In addition, being a small molecule, it was also free to diffuse within the silica polymer network, and its presence in the liquid phase retained by the silica network assisted in preventing the inorganic polymer from undergoing syneresis [33].

The results in vivo clearly demonstrate the usefulness of this formulation approach. Notably, the administration of SB3 in either a buffer or metolose-G did not improve diabetic ulcer healing, indicating the importance of providing a constant supply of fresh bioactive agent at the site of action by means of a slow release.

It is also noteworthy that the use of the silica–metolose-G made it possible to highlight the usefulness of SB3 in the treatment of ulcers. This result—which potentially brings a novel pharmacological option for chronic ulcers, with great value for patients—could not have emerged in the absence of a suitable formulation.

A system to deliver SB3 and boost fibrogenesis and angiogenesis will represent an invaluable tool to improve wound healing. Our study, however, has some limitations. First, the treatments were applied acutely, whereas repeated administrations would probably improve the performances of the wet sol–gel silica slurry. Second, we tested a single dose, extrapolated by the experiments in vitro. Dose-ranging experiments, coupled with repeated administrations, would identify the best combinations for improving the efficacy of our treatment, and will be the subject of future investigations.

With respect to a potential clinical translation, one could say that the formulation was here tested only immediately after its preparation, and it is expected that, upon shelf storage, the protein would diffuse out from the wet silica gel to the HPMC hydrogel environment, so that the formulation would lose its SB3 slow-releasing and protecting features. However, it has to be noted that the protein release from wet silica almost stops when the gel is stored in a silicic acid-saturated solution [18], a concentration that is reached quite fast when storing the wet gel in the minimum amount of buffer solution to keep it hydrated. As long as the hydration buffer is saturated with silicic acid, the protein will not diffuse out for the wet gel and the slurry will maintain its slow-releasing potential, which will be fully restored only upon dilution with the HPMC hydrogel. Therefore, a packaging solution capable of permitting the mixing of the two gels immediately prior to application should be devised to make this formulation approach translatable to the clinic.

## 5. Conclusions

The results of this work bring about two major findings: (a) for the first time, the potentials of Serpin B3 were demonstrated as a novel pharmacological option in the treatment of diabetic foot ulcers, a serious and disabling disease which, to date, does not have any specific treatment available, and (b) treatment with SB3 was effective in vivo only when the protein was formulated using the wet silica gel.

The wet sol–gel silica/HPMC approach may represent a new convenient formulation solution in the yet not fully explored field of protein delivery to the outer skin. In fact, in principle, its use may also be extended to the formulation of other bioactive proteins.

## 6. Patents

The patents #IT 102022000000884 and #IT102022000002618 are pending to the University of Padua.

## Figures and Tables

**Figure 1 pharmaceutics-14-01944-f001:**
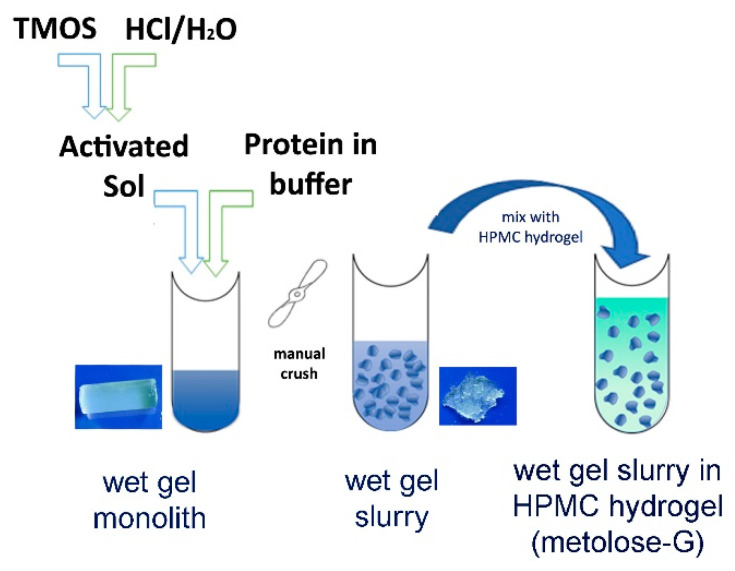
Scheme of formulation preparation, including protein entrapment in a wet sol–gel silica monolith, crushing into a wet silica slurry, and suspension of the silica slurry in the HPMC hydrogel. The images in the pictures are representative of a 2 mL monolith (about 1 × 2.5 cm) and of a slurry obtained from 0.5 g of monolith.

**Figure 2 pharmaceutics-14-01944-f002:**
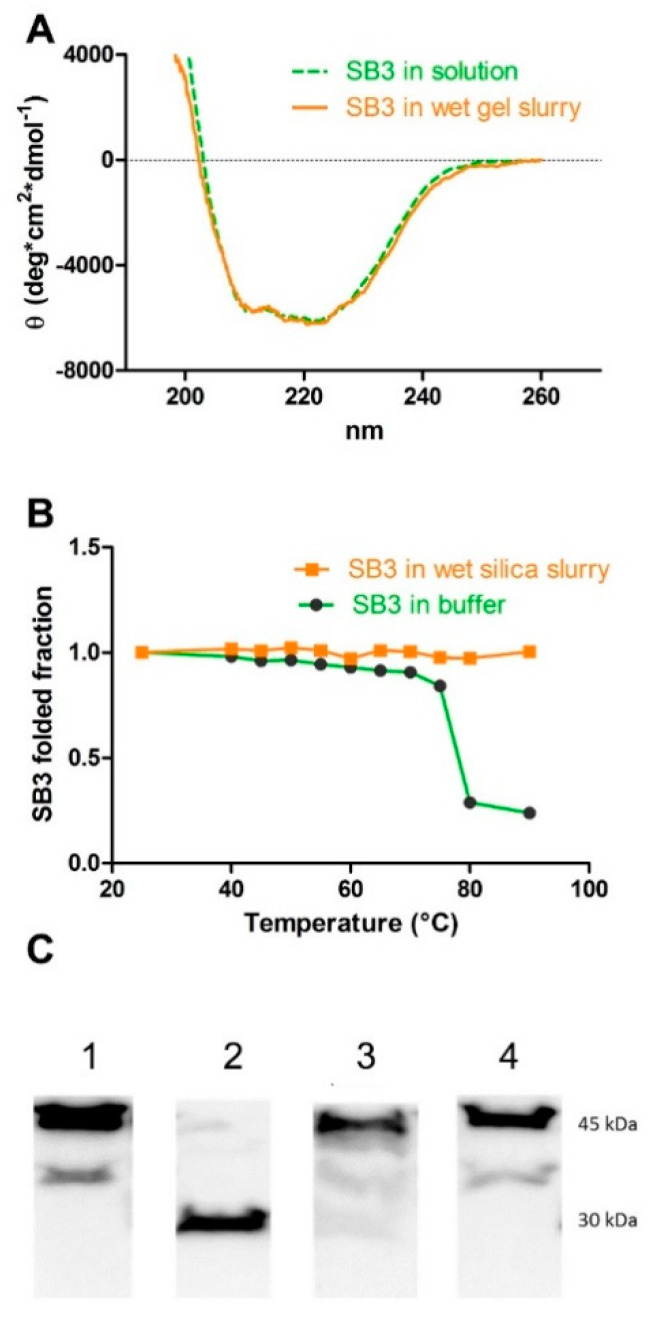
Effect of wet sol–gel silica entrapment on the conformation and stability of SB3. (**A**) Far UV circular dichroism spectra of SB3 registered at 25 °C in solution (full line) and in the gel slurry (dashed line). (**B**) Heat denaturation profile of SB3 registered at 25 °C in solution (circles) and in the gel slurry (squares). (**C**) Western blot of SB3 differently treated: (1) control SB3 in solution (2 µg/mL); (2) SB3 in solution after incubation (1 h, 37 °C) with pronase; (3) SB3 released from silica gel, previously incubated (1 h, 37 °C) with pronase; and (4) SB3 released from the silica slurry previously treated for 1 h at 37 °C with a buffer and without pronase. The band at 45 kDa corresponds to native SB3.

**Figure 3 pharmaceutics-14-01944-f003:**
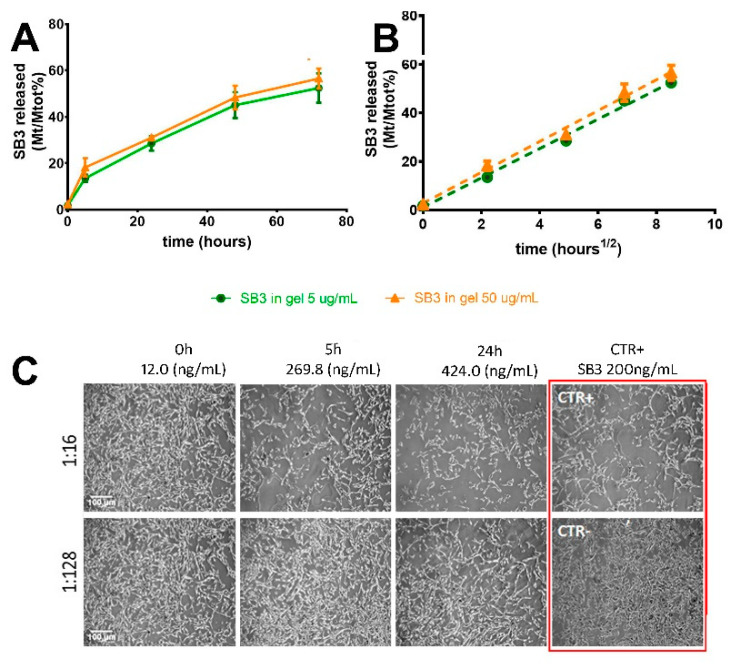
Release of SB3 from the 8% wet sol–gel silica slurry at 37 °C. (**A**) Mt/Mtot% SB3 as a function of time (h). (**B**) Mt/Mtot% SB3 as a function of the square root of time (h). Gels were loaded with the protein at an SB3 of 5 µg/mL (●) and 50 µg/mL (Δ). SB3 quantification in the release media was carried out using ELISA. (**C**) SB3 activity in the release solution from the gel at 50 µg of SB3/mL was assessed by the scatter assay on MDK cells. The release solution isolated after 0, 5, and 24 h of contact with the wet gel (37 °C) was added to the MDK cells at two dilutions (1:16 and 1:128). The scatter activity result of the release solutions was compared to that obtained with recombinant SB3 (200 ng/mL) used as a positive control (CTR+).

**Figure 4 pharmaceutics-14-01944-f004:**
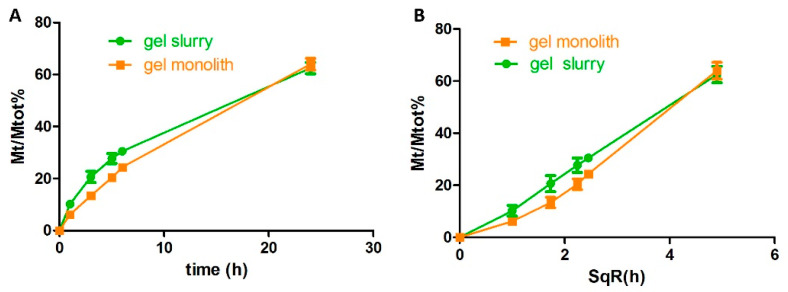
Release profiles of BSA–FITC from wet gel monoliths (squares) or silica slurries (circles). Release data are plotted as a function of time (**A**) and its square root (**B**).

**Figure 5 pharmaceutics-14-01944-f005:**
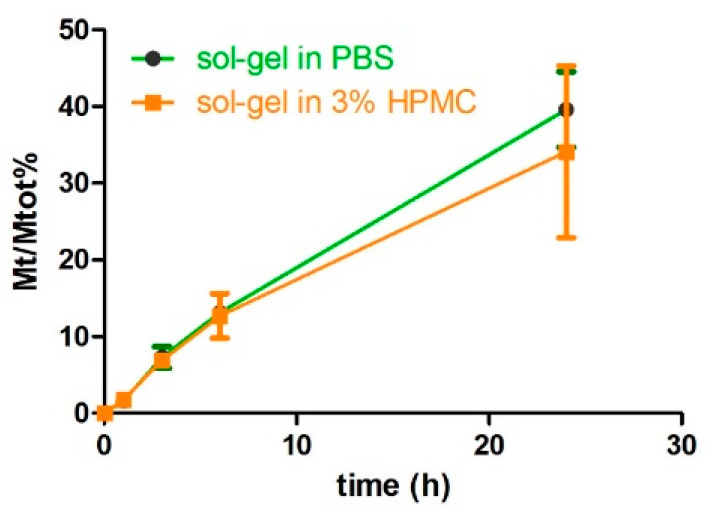
Release profiles of BSA–FITC (385 µg/mL) from silica slurries in the presence (○) and absence (●) of a 3% HPMC hydrogel. Note: The release assay was carried out in dialysis tubes (cut-off of 100 kDa) and this explains why the overall rates are lower than what is reported in Figure 2.

**Figure 6 pharmaceutics-14-01944-f006:**
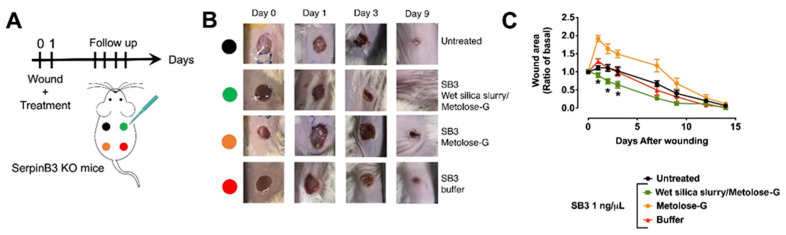
SB3 in wet silica gel slurry–metolose-G was effective at improving wound healing in serpin B3 KO mice. (**A**) Experimental layout for the inductions of the wounds, the application of treatments, and the assessment of healing. (**B**) Representative photographs of wounds at different time points. (**C**) Wound area of the different treatments, reported as the ratio of the healed and initial areas. * *p* < 0.05 for wet sol–gel silica compared to the other treatments (2-way ANOVA). N ≥ 7 for each group.

**Figure 7 pharmaceutics-14-01944-f007:**
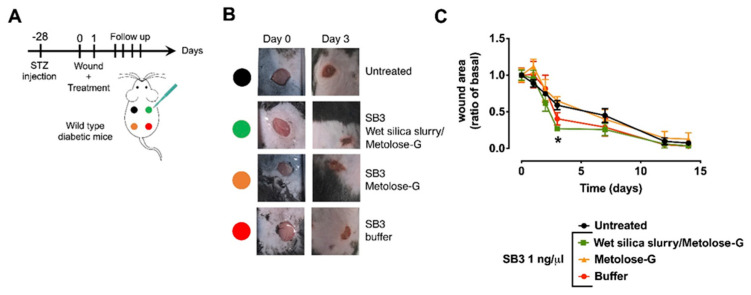
In vivo efficacy of SB3 formulations in the treatment of diabetic ulcers. (**A**) Experimental design for the induction of diabetes and the executio1n and treatment of the wound. (**B**) Representative images of diabetic ulcers at baseline time (0) and 3 days after injury. (**C**) Wound-healing profiles in animals treated with vehicles only (black line) and with SB3 formulated in the different vehicles (colored lines). * *p* < 0.05 after 2-way ANOVA.

## Data Availability

Not applicable.

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
