# Peer review of "Semisolid Wet Sol–Gel Silica/Hydroxypropyl Methyl Cellulose Formulation for Slow Release of Serpin B3 Promotes Wound Healing In Vivo"

_pharmaceutics, 2022, doi:10.3390/pharmaceutics14091944_

Round 1

Reviewer 1 Report

In general,  I accept after the minor revision cited below.

- Authors must correct the position of references in the introduction text.

- At various times, the authors mention the methodology in the results item. I suggest that the authors remove these phrases from the text. An example: Item 3.1. describes a methodology. The same was observed in 3.1.1. and 3.2.

- Check the number of topics.

Absence of item 3.3.

Repetition of item 3.4.

Author Response

Please see the attcahment

Reviewer 2 Report

In the present manuscript, the authors reported a novel study claiming that the protein SerpinB3 (SB3) was identified as a positive biomarker of successful diabetic wound healing, therefore, its exogenous administration may promote healing utilizing gel system for control release. The manuscript is well written however, there are few comments that needs to be addressed. 

The authors should pay more attention on the following points.

1.       Line 2, write HPMC in full.

2.       Line 58 and 59 remove – between sentences

3.       The in-text references should be before the full stop. E.g., Indeed, most of the strategies for the delivery of proteins described in the literature are finalized to parenteral application or local administration for the purpose of a transdermal or systemic effect [11]. Fix this throughout the manuscript and be consistence.

4.       Specify the SB3 percentage entrapped by the gel

5.       Figure 3 A and B, include more data points.

6.       Figure 3 C, indicate a scale bar

7.       In references section, abbreviate all journal names.

Reviewer 3 Report

In this manuscript, a sol-gel silica/hydroxypropyl methyl cellulose (HPMC) hydrogel blend containing SerpinB3 (SB3) was developed. The effects of silica entrapment on SB3 structure and biological activity and its release kinetics was evaluated. And two animal models were used to study the stimulant effect of a SB3 loaded formulation on wound healing. The following concerns should be addressed prior to acceptance.

1. In line 305-306, it was mentioned that "The results clearly demonstrate that, as long as SB3 is confined within 305 the wet silica gel network, it is protected from any proteolytic attack". To prove this conclusion, it is necessary to carry out the test in the presence of different enzymes Whether the conclusion reached on line 305 is too absolute. It is better to make some corrections.

2. Since this study was intended to show that the hydrogel should guarantee a supply of active agent for at least 3 days, why was the release test not directly conducted for this long time, but only for 1 day (Figure 3)?

3. The numbering of each section in the manuscript needs to be carefully proofread. For example, the section number of "3.1.1" is duplicated but that of "3.3" is missing.

4. It is mentioned in line 381 that “In mice, wound healing occurs via contraction, rather than epithelialization and, to overcome this obstacle, surgical dressings featuring semi-permeable membranes can be applied to the wounds.” The description is somewhat confusing. In many studies on wound healing of full-thickness skin wound on mice model, silicon splints are usually used instead of surgical dressings to minimize the effect of skin contraction on the wound healing. Surgical dressings were largely used to promote the wound healing.

5. Are there any statistically significant differences between the groups in Figure 6C and 7C?

Round 2

Reviewer 3 Report

This manuscript has been revised based on the reviewer's previous comments properly. It is acceptable in this form.